# Assessing Microbial Monitoring Methods for Challenging Environmental Strains and Cultures

**Damon C. Brown and Raymond J. Turner ***

Biological Sciences, University of Calgary, Calgary, AB T2N 1N4, Canada; browndc@ucalgary.ca
* Correspondence: turnerr@ucalgary.ca

**Abstract:** This paper focuses on the comparison of microbial biomass increase (cell culture growth) using field-relevant testing methods and moving away from colony counts. Challenges exist in exploring the antimicrobial growth of fastidious strains, poorly culturable bacteria and bacterial communities of environmental interest. Thus, various approaches have been explored to follow bacterial growth that can be efficient surrogates for classical optical density or colony-forming unit measurements. Here, six species grown in pure culture were monitored using optical density, ATP assays, DNA concentrations and 16S rRNA qPCR. Each of these methods have different advantages and disadvantages concerning the measurement of growth and activity in complex field samples. The species used as model systems for monitoring were: *Acetobacterium woodii*, *Bacillus subtilis*, *Desulfovibrio vulgaris*, *Geoalkalibacter subterraneus*, *Pseudomonas putida* and *Thauera aromatica*. All four techniques were found to successfully measure and detect cell biomass/activity differences, though the shape and accuracy of each technique varied between species. DNA concentrations were found to correlate the best with the other three assays (ATP, DNA concentrations and 16S rRNA-targeted qPCR) and provide the advantages of rapid extraction, consistency between replicates and the potential for downstream analysis. DNA concentrations were determined to be the best universal monitoring method for complex environmental samples.

**Keywords:** optical density (OD); ATP; DNA; 16S rRNA; qPCR; microbial growth; biomass; diverse species

## 1. Introduction

Assessing growth is fundamental to nearly all microbial studies. On the surface, this is an easy procedure carried out in introductory courses worldwide [1]. However, it turns out that this routine experiment is not as trivial as one thinks. For easily culturable aerobic species, the process is relatively simple as the growth medium need only contain the appropriate carbon sources and essential nutrients to culture the typical well-studied model microbes. After the appropriate growth medium has been selected, direct cell counting on agar plates can be performed for accurate quantification, providing colony-forming units (CFU) or viable cell count (VCC) values. However, it has become apparent that this method restricts the scope of species possible for study and cannot be used to study complex environments [2–4]. Here, we take the opportunity to review culture monitoring methods as a resource for readers and to provide context to our study.

For more rapid analysis, optical density (OD) evaluates the scattering of light by cells, either using the classical Klett meter or an absorption spectrometer set to 550 or 600 nm. Klett units are a similar means of determining cell concentrations using turbidity, where the turbidity of a liquid culture is correlated to a colony-forming unit value, and this is commonly conducted on a per-strain basis via wavelength filters as part of this older tool [5,6]. If a sample or culture has high turbidity, the effect of light scattering by the cells is diminished and the measured OD becomes too high to provide a linear application of the Beer–Lambert law. Studies have shown that OD measurements to

assess cell counts are highly dependent on the spectrophotometer, wavelength, media type, growth stage, cell morphology and the presence and concentration of secreted compounds. It is often overlooked that OD measurements should be taken as a proxy and not as concrete correlations [7]. The direct comparison of OD for different species is difficult due to the changes in turbidity resulting from the cell shape, such as rod compared to coccoid and from cell agglomeration to flocculation issues, and thus not from individual cell density, which can lead to incorrect assumptions at the same OD. This will not pose an issue while studying a pure culture of known shape, as comparisons are direct. However, for cross-comparisons or for mixed species/strain cultures, this becomes a significant issue and lowers the utility and accuracy of OD as a tool for growth measurement.

Other metabolic or growth-specific methods exist, such as dilution series (colloquially known as "bug bottles") [8] and bacteriological activity reaction test (BART) bottles [9], both of which produce a color change resulting from specific microbial growth and metabolism, where the time until detectable change (i.e., a change in the color of the media) is used to approximate the initial microbiological cell count. These are commercially available for sulfate-reducing bacteria, acid-producing bacteria, iron-reducing bacteria and others. All of these methods are dependent on the culturability of the microbes in question and are not applicable when the species in question cannot be cultured easily or conveniently. Due to the nature of these techniques and their estimation of culturable microorganisms and not quantitative enumeration, these techniques are more commonly used in industry, where exact cell counts are not required, as opposed to scientific endeavors, which typically require more precise counts.

Moving away from the culture-dependent methods, dyes and stains can be used to visualize and semi-quantitatively measure growth. Direct cell counting using microscopy can be used without (grid cell counting) and with staining [10–12]. Many stains exist to visualize microbial growth, such as crystal violet and safranin, which can be used for biofilm assessment [13,14], 5-(4,6-dichlorotriazinyl) aminofluorescein (DTAF) for total biomass staining [15,16], or metabolic dyes to determine actively respiring cells, such as 5-cyano-2,3-ditolyl tetrazolium chloride (CTC) [16]. These stains do not provide true cell counts, but produce quantifiable cell metabolic activity or biomass measurements, which, depending on the research question posed, is sufficient to monitor microbial growth.

Flow cytometry can be added to staining approaches to measure live and dead cell numbers [17], providing a rapid (minutes) method to count cells compared to conventional plate counting (days) and can measure cell counts in the $10^{2-7}$ CFU/mL range [18]. It requires staining of the bacterial DNA using ethidium bromide, SYTO-9, hexidium iodide or staining for other bacterial components such as the cell wall using Oregon Green® conjugated wheat germ agglutinin [19]. Flow cytometry also offers the ability to sort cells based on labeling, provided the required equipment is available [20]. This approach is elegant, but is certainly still not routinely available to most labs or in a field setting, and may be unnecessary for the majority of questions being asked.

Fluorescent in situ hybridization (FISH) is a target-specific approach that relies on a fluorescent reporter attached to a nucleic probe to determine the presence and abundance of the target sequence. This can be used for total or genera-specific cell enumeration when targeting a gene such as 16S rRNA [21,22]. Now, in our genomics era, 16S rRNA quantification using quantitative PCR (qPCR) is gaining popularity with [23] or without sequencing databases [24,25].

For anaerobes, accurate cell enumeration becomes increasingly more difficult as different anaerobes will require different oxidation-reduction potentials (ORP) to thrive; $\leq -100$ mV for obligate anaerobes [26] and $\leq -330$ mV for strict anaerobes [27]. Pure-culture anaerobes can be cultured and enumerated using improved Hungate culturing techniques [28–31], but direct microscopy and FISH are more common [32–35]. A review of anaerobic culturing and quantification is available elsewhere [36]. In many instances, the difficulty of culturing anaerobes forces researchers to choose indirect methods to assess growth and activity, such as rates of substrate consumption or end-product production,

where an easily quantified chemical is sampled and measured at time points in favor of actual cell counting [37,38]. The rate of consumption or production relates to cell growth through the Monod equation [39]. It should be noted that there is a distinction between cell counting and microbial activity, and the need to monitor one, the other or both depends on the research, environmental, industrial or medical question being asked.

Other, less direct methods exist, which can approximate cell counts through the quantification of other components such as key metabolites including ATP or major biochemical compositions such as proteins, DNA or lipids. The choice of brand or instrumentation used to measure any of these components is inconsequential as long as the choice is consistent. This eliminates the impact of systematic error present in all analytical approaches, and allows monitoring of the trends of the chosen biological component. Each of these techniques relies on an average quantity of the target molecule being present in each cell, and ignores the potential fluctuation of concentrations resulting from changes during a specific stage of cellular division [40–42]. Lipids can be used to determine growth rates by tracking the incorporation of heavy water using gas chromatography [43]. In the case of ATP, assays use an average quantity of ATP per cell based on Escherichia coli, where one E. coli cell contains approximately 1 femtogram of ATP [44]. Studies have shown that ATP concentrations are stable throughout all growth rates, although exact ATP concentrations per cell were not calculated [45]. These assumptions do not consider periods of external stress (e.g., biocide exposure) or temperature increases, which may increase the intracellular ATP concentrations in response [46]. An added benefit of these methods, when considering non-defined environmental samples, is that they are only present in biochemically active cells, and free molecules do not survive for long outside of a living cell. As such, in environmental samples, these lines of evidence can be reasonably used to estimate the total biomass without having to consider the types or diversity of species present.

Advanced molecular methods such as PCR can quantify specific gene copy numbers including 16S rRNA genes, housekeeping genes or other species-specific marker genes to enumerate the organisms present in complex samples. Many databanks exist that have developed different 16S rRNA primer sets, each with their own biases towards detecting or omitting certain microbial clades and are reviewed elsewhere [47–49]. While 16S rRNA is a universal gene target and has been used dating back to 1999 [50], other housekeeping genes specific to a species of interest may be used to obtain cell counts of targeted populations. For example, the use of primers targeting dissimilatory sulfate reductase, *dsrA,* to monitor sulfate-reducing organisms [51] or nitrite reductase, *nirS,* to quantify *Pseudomonas stutzeri* [52] or metabolic potential e.g., aromatic oxygenases [53] and hydrocarbon hydroxylases [54] for the bioremediation potential of an environment. An important consideration when using genes as a cell count proxy is the copy number of the gene in each cell. For 16S rRNA specifically, bacteria can have anywhere from a single copy up to 15, with an average of $3.82 \pm 2.61$ [55]. These primers can be used in quantitative PCR (qPCR) for total cell counts, or in sequencing to acquire relative abundance values. Sequencing can be targeted as in the case of a 16S rRNA community profile [56–58] or a whole metagenome to determine the diversity of genes and/or taxa present [59–61].

The above exemplifies the issues of accurate quantification of aerobic and anaerobic pure cultures. This task becomes exponentially more difficult when considering environmental samples that have diverse, unspecified microbial species present. This is a wide-reaching problem, occurring in industries from agriculture [62], oil and gas [63], clinical [64], wastewater treatment [65], hydrocarbon bioremediation [66], cosmetics [67] and food packaging [68]. In such examples, highly specific monitoring methods are no longer viable to determine accurate cell counts; thus, general techniques must be used. The typical trade-off when transitioning from selective to general monitoring is the loss of specificity (e.g., the presence of a specific pathogen) for the gain of total cell counts.

A brief summary of the main cell-monitoring methods and relative costs is provided in Table 1. These relative costs aim to provide a scale of cost for the equipment and necessary reagents for each method, while listing the advantages and disadvantages of

each approach to better provide direction for which method is best suited for a particular question and/or scenario.

**Table 1.** Comparison of advantages, disadvantages and relative costs of common enumeration techniques.

| Method | Advantages | Disadvantages/Limitations | Relative Cost [a] |
|---|---|---|---|
| Culturing | -Easy to quantify <br> -Easy to determine presence on contamination | -Unable to quantify complex communities <br> -Very limited number of species can be grown <br> -Time required depends on species' doubling time | $10 |
| Optical density | -Fast <br> -Sample is recoverable | -True quantification is limited by transmission <br> -Counts should be verified by another method <br> -Precipitants in liquid interfere | $1 |
| Staining and microscopy | -Combining stains allows differential counting | -Biased by field of view in microscope <br> -Dense growth prevents accurate counting | $100 |
| ATP assay | -Rapid test able to be used in lab or in the field <br> -Measures ATP from all living cells | -One-time equipment is expensive, reagents are consumable and expensive <br> -Microbial equivalent calculations do not provide true cell count | $1000 |
| DNA concentration | -DNA can be used for downstream applications <br> -Once extracted, DNA is stable for long time frames, allowing flexibility in measuring | -Minimum volume/cell mass of sample required <br> -Variance in DNA concentrations between species skews cell count conversions <br> -DNA must be extracted then measured | $100 |
| Flow cytometry | -Very accurate enumeration <br> -Combining different stains enables counting of multiple types of cells | -Requires expensive equipment <br> -Not applicable to environmental samples | $1000 |
| Quantitative PCR | -Enumeration can be as broad or specific as desired based on primers used <br> -Multiplexing can count multiple targets in single runs | -Expensive equipment and consumable reagents <br> -Technically more challenging than other methods | $1000 |
| Metagenomic sequencing | -Detects all species present in complex environments | -Provides relative cell counts, not true cell counts <br> -Equipment is very expensive | $1000–10,000 [b] |

[a] Values do not consider the cost of instrumentation. Thus, the initial start-up cost can vary considerably, even within a given method depending on the features. For plate counting, this can all be robotics and automated optics. Costs vary over 2–3-fold for most instrumental tools. [b] Many services and companies exist to run samples and reduce the cost of reagents and equipment time, eliminating the cost of equipment.

In this study, we compare selected testing methods used in the laboratory settings of academic/environmental/industrial applications using six pure cultures as model systems. We omit the use of field samples, as their growth measurement would not provide information regarding how the different monitoring methods relate to each other, whereas we can expect sigmoidal growth curves from pure cultures. From our group's interest, we will look at ATP levels and 16S rRNA qPCR, which are being adapted in oil and gas industries in western Canada, and compare them with traditional $OD_{600}$ and DNA concentrations to determine how reliable and complementary these differing methods are. This work aims to help bridge the gap between pure cultures studied in laboratory settings, where there are many different enumeration methods, and situations where there is not a clear and obvious enumeration method to deal with complex samples. By applying these techniques to different pure cultures, we explore how well the methods agree with each other for each of the species and between species to understand how these fundamental microbial methods compare and are affected by different microbes. Here, by monitoring the microbial biomolecules and metabolic activity with time, we can assess how well each monitoring approach is able to capture subtleties in the curve shapes.

## 2. Materials and Methods

### 2.1. Cultures and Media

The six pure cultures used in this study were acquired from DSMZ (Deutsche Sammlung von Mikroorganismen und Zellkulturen, Braunschweig, Germany). The cultures are *Acetobacterium woodii* (DSM 1030), *Bacillus subtilis* (DSM 10), *Desulfovibrio vulgaris* (DSM 644), *Geoalkalibacter subterraneus* (DSM 29995), *Pseudomonas putida* (DSM 291) and *Thauera*

*aromatica* (DSM 6984). Each of the chosen species has a representative genome sequenced on NCBI and their details are in Table 2. The pure cultures were recovered from $-70\ ^\circ$C freezer stocks (10% glycerol) into the suggested medium prepared in 20 mL aliquots and sealed in 26 mL Hungate tubes with anaerobic headspaces (either $N_2$ or $N_2/CO_2$). The medium was boiled and purged with anaerobic gas. To promote growth, the media tubes for *B. subtilis* and *P. putida* had a headspace of 4 mL air injected through a 0.2 μm filter. Fresh media tubes were inoculated with the freezer recovery culture and incubated at the recommended temperatures for 7 days to ensure a stationary phase was reached, and then fresh media tubes were inoculated in triplicate with a 10% by volume inoculant and used in testing. The growth conditions for each species are listed in Table 3.

**Table 2.** Representative sequenced genome chosen and identified 16S rRNA copy numbers.

| Species and Strain | NCBI Accession Number | Genome Length (bp) | 16S rRNA Copy Number * |
|---|---|---|---|
| *Acetobacterium woodii* DSM 1030 | CP002987.1 | 4,044,777 | 5 |
| *Bacillus subtilis subtilis* 168 | CP010052 | 4,215,619 | 10 |
| *Desulfovibrio vulgaris* Miyazaki F | CP001197.1 | 4,040,304 | 4 |
| *Geoalkalibacter subterraneus* Red1 | CP010311 | 3,475,523 | 4 |
| *Pseudomonas putida* KT2440 | AE015451.2 | 6,181,873 | 7 |
| *Thauera aromatica* MZ1T | CP001281.2 | 4,496,212 | 4 |

* Counted manually from NCBI sequences.

**Table 3.** Growth conditions for each species.

| Species and Strain | Medium | Temperature (°C) |
|---|---|---|
| *Acetobacterium woodii* DSM 1030 | DSM 135 | 30 |
| *Bacillus subtilis subtilis* 168 | DSM 1 | 30 |
| *Desulfovibrio vulgaris* Miyazaki F | DSM 63 | 37 |
| *Geoalkalibacter subterraneus* Red1 | DSM 1249 | 37 |
| *Pseudomonas putida* KT2440 | DSM 1a | 26 |
| *Thauera aromatica* MZ1T | DSM 586 | 30 |

### 2.2. Sampling and Testing

Time points for each species were decided upon based on the preliminary growth experiments and previous knowledge of each organism (see Supplementary Materials). During these screens, the length of the lag phase and beginning of the stationary phase were identified, and subsequent time points were chosen to encompass these points. Two to three time points were chosen to cover the lag phase, logarithmic growth phase and the stationary phase.

Once inoculated, the fresh cultures were sampled in a time course for testing. At each point, sterile syringes and needles were used to aseptically remove 2 mL. Each biological replicate was used as a single replicate for each method (totaling three replicates), with technical duplicates used in qPCR (three biological replicates with two technical replicates each). This aliquot was split, with 1 mL used to measure $OD_{600}$ in a UV-Visible Spectrophotometer (Hitachi U-2000, Hitachi, Chiyoda, Japan) using uninoculated media as the reference sample. The sample was then recovered and used for DNA extraction in the FastDNA® Spin Kit (MPBio, Irvine, CA, USA). The DNA concentration was measured using a Qubit™ fluorometer (Invitrogen, Carlsbad, CA, USA) and the Quant-iT™ dsDNA HS assay kit (Thermo Scientific, Waltham, MA, USA). Following quantification, the DNA was cleaned using the OneStep™ PCR inhibitor removal kit (Zymo Research, Irvine, CA, USA) prior to being used in qPCR. The other 1 mL was consumed to measure ATP using the LifeCheck ATP test kits (OSP, Calgary, AB, Canada).

### 2.3. Quantitative PCR

qPCR was carried out, targeting the 16S rRNA gene, specifically 515–809 (variable region 4) using modified primers from Caporaso et al. [69]. The primer sequences are

provided in Table 4. The starting quantification of the 16S rRNA gene was determined using synthetic gBlocks purchased from IDT (Integrated DNA Technologies, Newark, NJ, USA) at concentrations of $10^{8-3}$ copies/µL in ten-fold serial dilutions. A standard curve was created using the Cq values from the gBlocks and used to calculate the starting quantities of the samples. Melt curve analysis was performed to ensure that all amplification was the result of intended binding and not non-specific binding. The gBlock used contained the target 16S rRNA sequence flanked by two multidrug resistance genes (for use in other studies), separated by sequences of 10 thymines. The entire gBlock sequence is provided in Table 5. Reaction mixtures were prepared to a total volume of 20 µL, with 10 µL PowerUp™ SYBR™ Green 2× Master Mix (Applied Biosystems, Waltham, MA, USA), 1.2 µL 10 µM 515_F, 0.6 µL 10 µM 806_R, 6.2 µL nuclease-free water and 2 µL DNA template. Thermocycling was performed in a CFX96 Real-Time PCR System (BioRad, Hercules, CA, USA) with the following protocol: 50 °C—2 min, 95 °C—2 min followed by 50 cycles of 95 °C for 45 s, 55 °C for 30 s and 72 °C for 45 s, and then a melt curve analysis was performed from 60–95 °C.

**Table 4.** Primer sequences used in quantitative PCR.

| Primer Name | Sequence (5′-3′) | Melting Temperature (°C) |
|---|---|---|
| 519_F | CAG **C**M**G** CCG CGG TAA | 57.6 |
| 806_R | GGA CTA **C**H**V** GGG T**W**T CTA AT | 50.7 |

Nucleotide codes: **H** = A/C/T; **M** = A/C; **V** = A/C/G; **W** = A/T.

**Table 5.** gBlock DNA sequence.

| gBlock DNA Sequence (5′-3′) (Multidrug Resistance Efflux Pump Gene A–(Thymine Spacer)–Universal 16S rRNA–(Thymine Spacer)–Multidrug Resistance Efflux Pump Gene A) |
|---|
| Multidrug resistance gene A amplicon TTTTTTTTTTT GTG CCA GCA GCC GCG GTA ATA CAG AGG GTG CAA GCG TTA ATC GGA ATT ACT GGG CGT AAA GCG CGC GTA GGT GGT TTG TTA AGT TGG ATG TGA AAG CCC CGG GCT CAA CCT GGG AAC TGC ATC CAA AAC TGG CAA GCT AGA GTA CGG TAG AGG GTG GTG GAA TTT CCT GTG TAG CGG TGA AAT GCG TAG ATA TAG GAA GGA ACA CCA GTG GCG AAG GCG ACC ACC TGG ACT GAT ACT GAC ACT GAG GTG CGA AAG CGT GGG GAG CAA ACA GGA TTA GAT ACC CTG GTA GTC C TTTTTTTTTT Multidrug resistance gene B amplicon |

### 2.4. Cell Count Calculations

Optical density is the simplest method for measuring microbial growth and many formulae have been put forth to convert OD to CFU/mL to reduce the need to perform continuous plate verification [70–72]. $OD_{600}$ was converted to cell count equivalents using a formula created by Kim et al. (2012) [73] studying *Pseudomonas aeruginosa*, which found the following relation of OD to colony-forming units (CFU), as shown in Equation (1):

$$\text{Colony-forming units (CFU/mL)} = 2 \times 10^8 \times \text{OD} + 4 \times 10^6 \tag{1}$$

It is noted that this formula should be confirmed by plating cells at unique OD wavelengths and values and validating the formula for each pure strain, as cell size and shape influence the light scattering from the cells. It is acknowledged that the differences in culture turbidity of the six species used in this study and the inability to grow all six on plated media means that converting $OD_{600}$ to CFU values will be an inaccurate conversion, but this was carried out to maintain uniformity between the datasets and to highlight the issue of using such conversion factors without confirming and modifying the equation empirically for each species.

The ATP measurements were converted into microbial equivalents (ME), according to the manufacturer's protocol, using the relative light units (RLU) from the manufacturer's luminometer as in Equation (2), where blank RLU is the extraction solution with the luciferase enzyme and standard RLU is a solution with known ATP concentration and the luciferase enzyme.

$$\text{Microbial equivalents (ME/mL)} = \frac{\text{(sample RLU} - \text{blank RLU)}}{\text{standard RLU}} \times \frac{10,000}{\text{Sample size (1 mL)}} \times 1000 \qquad (2)$$

The DNA concentrations were converted into two cell-count proxies. One used the assumption of 2 fg of DNA per cell based on the average 1.6–2.4 fg DNA/cell [74], and the other used the genome length and Equation (3).

$$\text{Cell count approximation (cells/mL)} = \frac{\text{DNA} \left( \frac{\mu g}{mL} \right) \times \text{Avogadro's number} \times \left( \frac{1\,\mu g}{0.000001\,g} \right)}{\text{genome length (bp)} \times 650\,g/mol} \qquad (3)$$

This equation uses the genome lengths provided in Table 2, omits any plasmids from consideration and assumes the molecular weight of a base pair to be 650 g/mol. The results of these two calculations are reported in Table 6 for selected time points for each species.

**Table 6.** Linear correlation * values determined from scatter plots of each growth-monitoring technique.

| Species | OD$_{600}$ vs. ATP | OD$_{600}$ vs. DNA | OD$_{600}$ vs. 16S | ATP vs. DNA | ATP vs. 16S | DNA vs. 16S |
|---|---|---|---|---|---|---|
| *A. woodii* | 0.97 | 0.96 | 0.98 | 0.92 | 0.93 | 1.00 |
| *B. subtilis* | 0.89 | 0.84 | 0.86 | 0.64 | 0.66 | 0.98 |
| *D. vulgaris* | 0.65 | 0.93 | 0.74 | 0.74 | 0.50 | 0.62 |
| *G. subterraneus* | 0.24 | 0.99 | 0.98 | 0.16 | 0.15 | 1.00 |
| *P. putida* | 0.98 | 0.99 | 0.71 | 0.99 | 0.73 | 0.77 |
| *T. aromatica* | 0.97 | 0.90 | 0.95 | 0.89 | 0.92 | 0.98 |
| Average | 0.78 | 0.94 | 0.87 | 0.72 | 0.65 | 0.89 |

* Reported as the R value from XY scatter plots of each data set against each other (see Supplemental Data Figures S3–S8).

To convert the qPCR values into cell counts, the copies of 16S rRNA genes per μL were converted to copies mL$^{-1}$, then divided by the 16S rRNA gene copies counted in the NCBI-sequenced genomes (see Table 2).

We selected a time point for each species to represent inoculation (time zero), mid log phase and stationary phase, and then used the above conversions to obtain the cell count equivalents, which are reported in Supplemental data, Table S2. The DNA cell count approximation was done using the average DNA concentration, as the diverse populations in field samples mean that exact conversions using genome lengths is impossible. The standard deviation values at each point were subjected to the same equations as the averages, and were not recalculated using the cell count values of each replicate.

To determine how closely the different methods agreed with each other, scatter plots of the data points from OD$_{600}$, ATP (ME/mL), DNA concentration (μg/mL) and 16S rRNA (copies/μL) were created, a linear correlation was determined for each pairing and the R value was calculated. The average values for the two given methods were compared at all time points and the linear correlation was calculated from the resulting scatter plots. This was repeated to compare each method with each species, and the reported R values are presented in Table 6. The closer the R values of the linear correlations are to 1.0, the stronger the two methods agree with each other.

### 2.5. Mixed Culture Testing

To validate this work on more complex environments, the four growth-monitoring methods were tested against a model microbially influenced corrosion community of four species (*D. vulgaris*, *G. subterraneus*, *P. putida* and *T. aromatica*). The mixed community was grown in a CDC Biofilm Reactor (CBR) (BioSurface Technologies Corporation, Bozeman, MT, USA) to be able to monitor both sessile and planktonic cells as a proxy for environmental samples. Briefly, the CBRs were connected in a closed-loop system to a shared reservoir containing artificial sea water media, where planktonic cultures of *D. vulgaris*, *G. subterraneus*, *P. putida* and *T. aromatica* were inoculated with a 5% vol. inoculum in a staggered fashion. As with pure-culture experiments, each strain was grown for seven days prior to being inoculated together.

The CBRs contained six vertically mounted coupon holders, each with three carbon steel coupons and a central baffle with a stir bar attached and set to 130 RPM using an electronic stir plate. The CBRs had anaerobic gas (10% $CO_2$/90% $N_2$) supplied to the headspace of both bioreactors to prevent air ingress and to maintain an anaerobic atmosphere. Media was pumped between the reservoir and bioreactors through a peristatic pump at a rate of 3.5 mL/min feeding into the top of each CBR, and a return line connected the effluent of the CBR back to the reservoir. Planktonic samples were collected from three-way valves mounted immediately downstream of the bioreactors in the tubing connecting the bioreactors back to the reservoirs.

The planktonic samples were collected one hour after the final inoculation to verify microbial activity. Seven days after the final inoculation, the planktonic samples were collected and tested as described above. Sessile cells were collected on day 7 by removing a single coupon sleeve from each bioreactor. The coupons were individually placed into the wells of a 12-well microtiter plate containing 2 mL of sterile media and were sonicated for 10 min (5 min with each side face up). The coupons were then removed and the 2 mL sonicated media used in testing as described in Section 2.2.

After sampling on day 7, the single media reservoir was removed and replaced with two reservoirs and the flow of the CBRs isolated from each other. One CBR was exposed to a low concentration of a biocide (either 1 ppm benzalkonium chloride, BAC, or 37.5 ppm tetrakis (hydroxymethyl) phosphonium sulfate, THPS) to ensure the monitoring methods could measure the resulting fluctuations in growth due to biocide exposure as an artificial stressor. Planktonic and sessile samples were taken on days 8, 10 and 14 of the experiment (corresponding to 1, 3 and 7 days of biocide exposure, respectively). After sampling on day 14, fresh media was flushed through both of the CBRs separately to remove the biocide (also performed with the CBR without the biocide). After flushing both CBRs, fresh media was added to the separate reservoirs and the flow continued for another seven days (to day 21) when sessile and planktonic samples were taken a final time.

The linear correlation values were calculated for each sample type (planktonic and sessile cells from CBR exposed to THPS and not exposed) and are reported in Table 7, and the samples from the BAC trial are reported in Table 8

**Table 7.** Linear correlation * values determined from scatter plots of the four sample types collected during CBR trials when exposed to 37.5 ppm THPS.

| Sample | OD$_{600}$ vs. ATP | OD$_{600}$ vs. DNA | OD$_{600}$ vs. 16S | ATP vs. DNA | ATP vs. 16S | DNA vs. 16S |
|---|---|---|---|---|---|---|
| Planktonic growth with THPS | 0.27 | 0.45 | 0.20 | 0.48 | 0.82 | 0.27 |
| Planktonic growth without THPS | 0.27 | 0.45 | 0.38 | 0.11 | 0.50 | 0.66 |
| Sessile growth with THPS | 0.04 | 0.83 | 0.27 | 0.40 | 0.33 | 0.39 |
| Sessile growth without THPS | 0.93 | 0.96 | 0.80 | 0.96 | 0.88 | 0.91 |
| Average | 0.38 | 0.67 | 0.41 | 0.49 | 0.63 | 0.56 |

* Reported as the R value from XY scatter plots of each data set against each other (see Supplemental Data Figures S9–S12).

**Table 8.** Linear correlation * values determined from scatter plots of the four sample types collected during CBR trials when exposed to 1 ppm BAC.

| Sample | OD$_{600}$ vs. ATP | OD$_{600}$ vs. DNA | OD$_{600}$ vs. 16S | ATP vs. DNA | ATP vs. 16S | DNA vs. 16S |
|---|---|---|---|---|---|---|
| Planktonic growth with BAC | 0.52 | 0.33 | 0.36 | 0.12 | 0.02 | 0.76 |
| Planktonic growth without BAC | 0.51 | 0.03 | 0.17 | 0.79 | 0.73 | 0.86 |
| Sessile growth with BAC | 0.20 | 0.70 | 0.77 | 0.34 | 0.51 | 0.96 |
| Sessile growth without BAC | 0.77 | 0.89 | 0.88 | 0.48 | 0.46 | 0.76 |
| Average | 0.50 | 0.49 | 0.55 | 0.43 | 0.43 | 0.84 |

* Reported as the R value from XY scatter plots of each data set against each other (see Supplemental Data Figures S13–S16).

## 3. Results

The results for the monitoring of the six pure cultures are shown in Figures 1–6. In each case, the time course of the species was monitored using OD$_{600}$, ATP (ME/mL), DNA

(µg/mL) and 16S rRNA-targeted qPCR. The results of the ATP monitoring were calculated into microbial equivalents (ME/mL). Each of the three biological replicates collected at each time point were run in duplicate on the qPCR thermocycler, raising the trial's N value to six for this line of monitoring. Error bars are the calculated standard deviation of the three biological replicates (six for qPCR). As our focus here is the comparison of the methods and the trends therein, the data from ATP and qPCR are not displayed in log scale in order to better illustrate the fluctuations, though the graphs in logarithmic scale are available in Supplemental Data Figures S1 and S2, respectively.

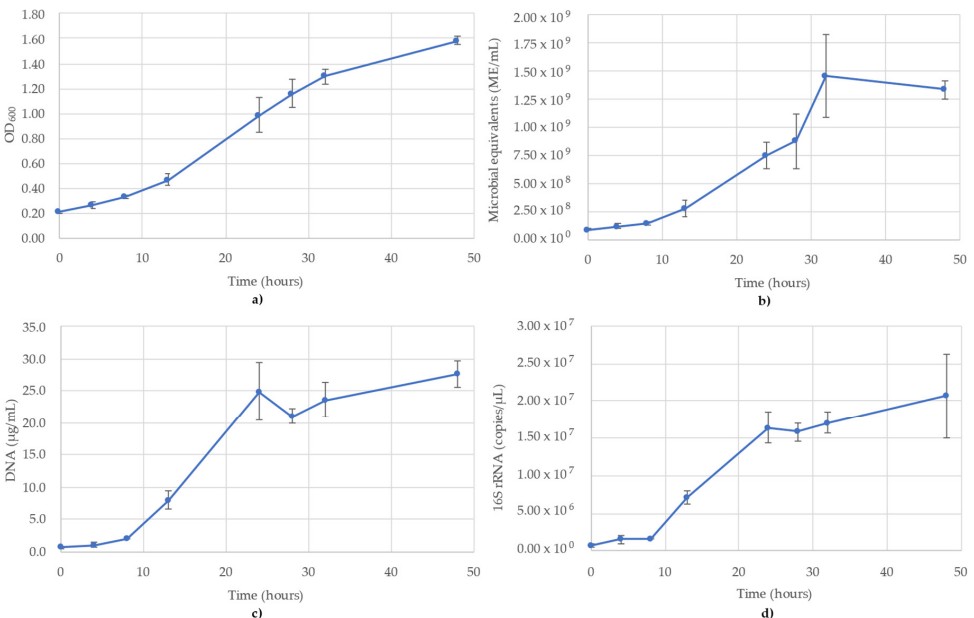

**Figure 1.** Monitoring of *A. woodii* using (**a**) $OD_{600}$; (**b**) ATP (ME/mL); (**c**) DNA concentration (µg/mL); (**d**) 16S rRNA targeted qPCR. $R^2$ value for the 16S rRNA qPCR standard curve was 0.987. Blue lines are the species growth curves (*n* = 3, for qPCR *n* = 6).

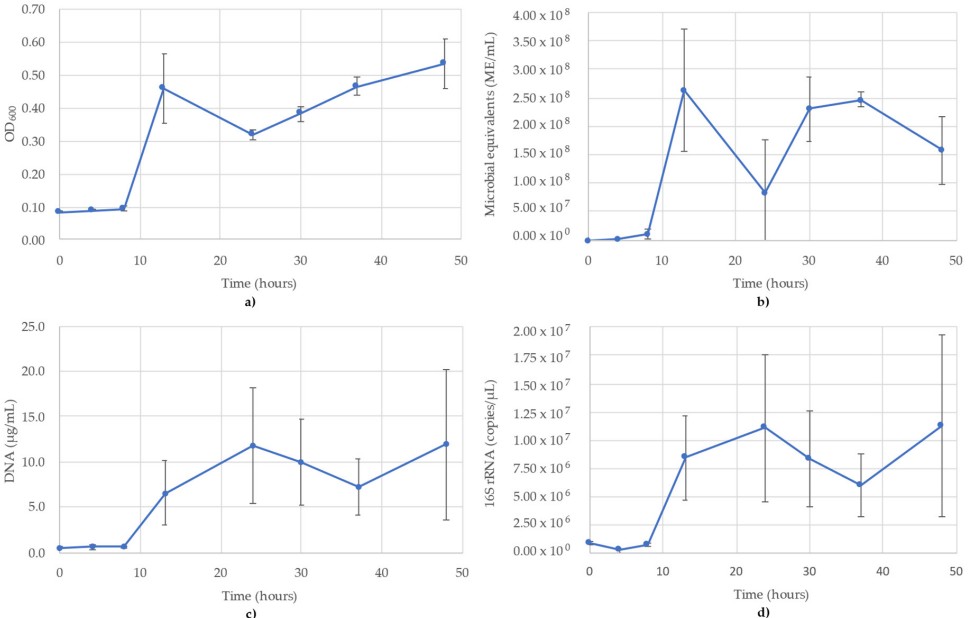

**Figure 2.** Monitoring of *B. subtilis* using (**a**) $OD_{600}$; (**b**) ATP (ME/mL); (**c**) DNA concentration (µg/mL); (**d**) 16S rRNA-targeted qPCR. $R^2$ value for the 16S rRNA qPCR standard curve was 0.995. Blue lines are the species growth curves (*n* = 3, for qPCR *n* = 6).

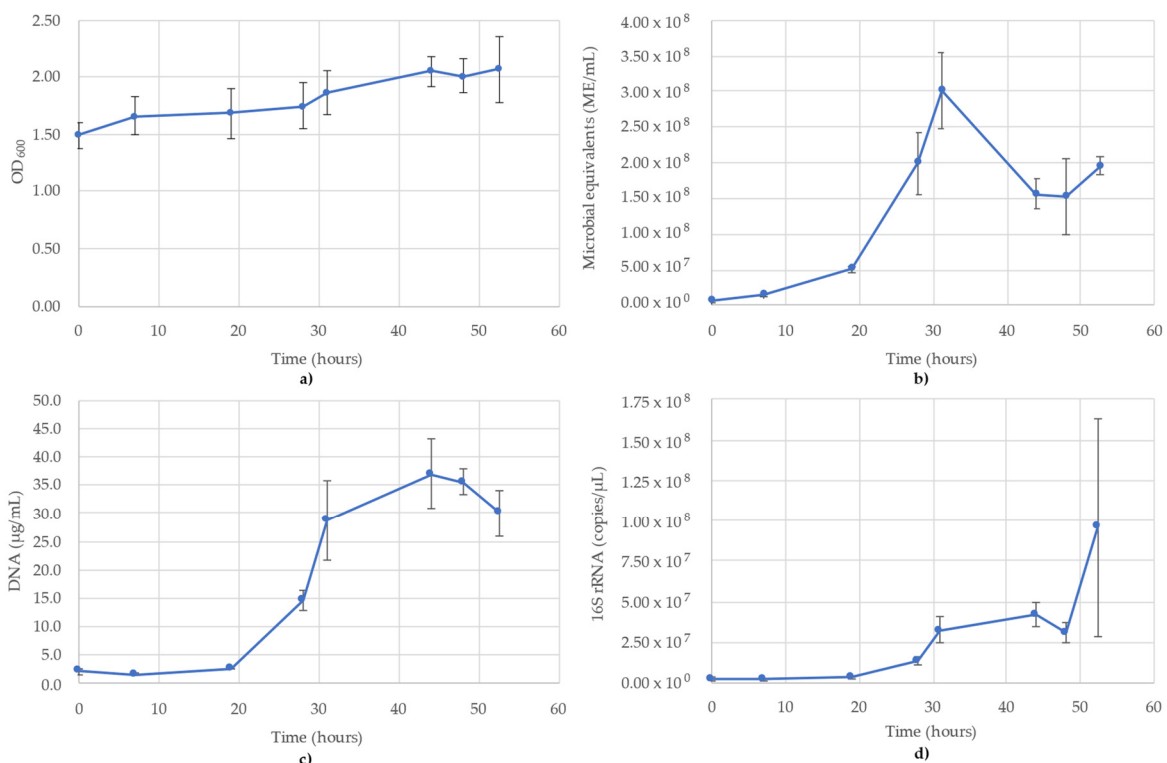

**Figure 3.** Monitoring of *D. vulgaris* using (**a**) OD$_{600}$; (**b**) ATP (ME/mL); (**c**) DNA concentration (μg/mL); (**d**) 16S rRNA-targeted qPCR. $R^2$ value for the 16S rRNA qPCR standard curve was 0.993. Blue lines are the species growth curves (*n* = 3, for qPCR *n* = 6).

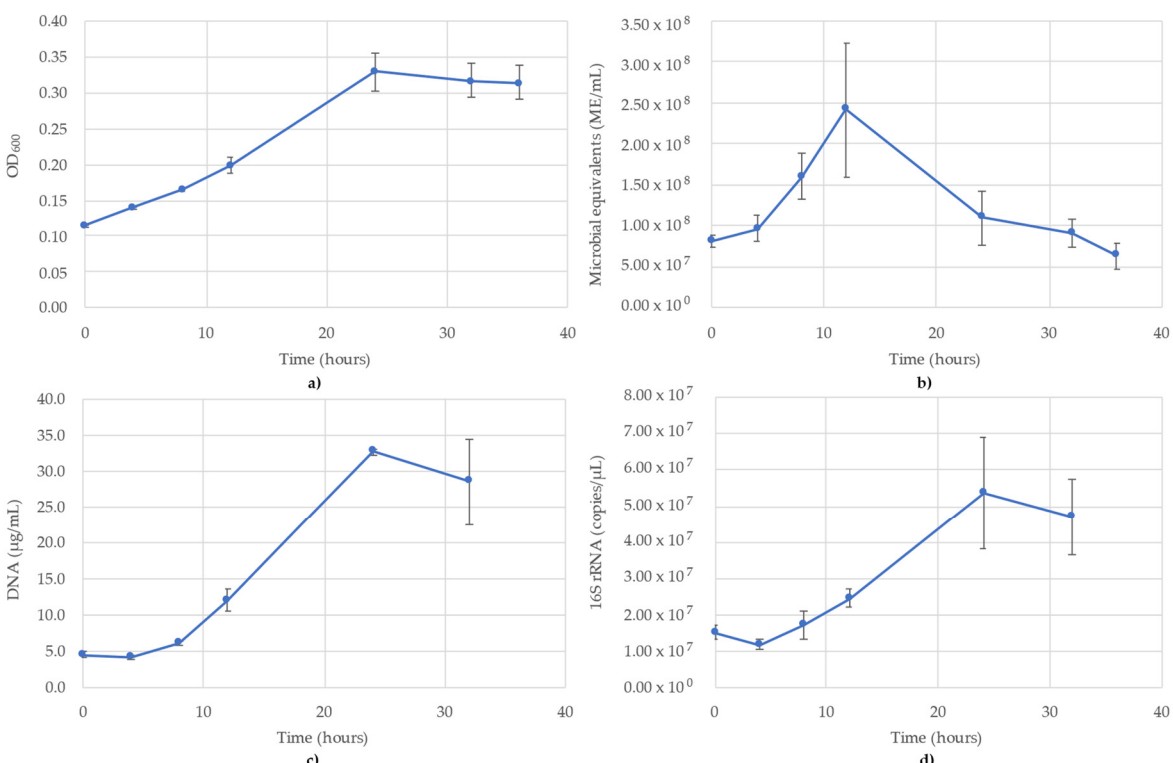

**Figure 4.** Monitoring of *G. subterraneus* using (**a**) OD$_{600}$; (**b**) ATP (ME/mL); (**c**) DNA concentration (μg/mL); (**d**) 16S rRNA targeted qPCR. $R^2$ value for the 16S rRNA qPCR standard curve was 0.961. Blue lines are the species growth curves (*n* = 3, for qPCR *n* = 6).

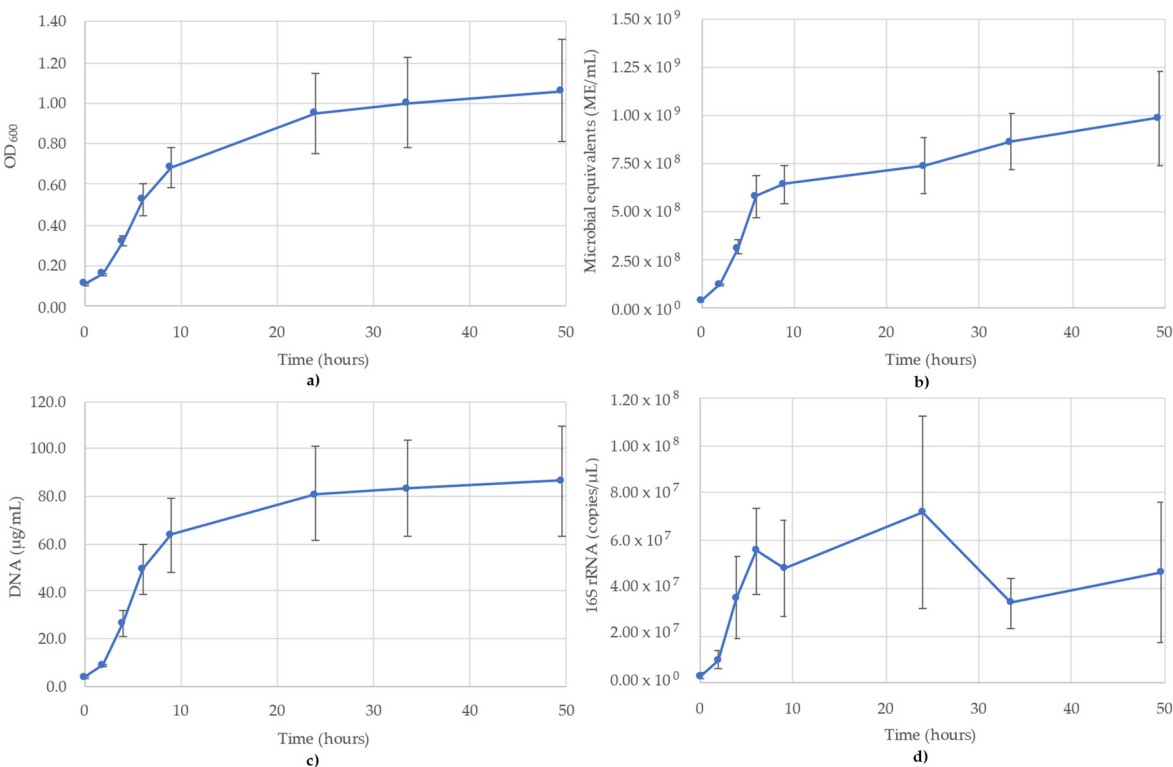

**Figure 5.** Monitoring of *P. putida* using (**a**) OD$_{600}$; (**b**) ATP (ME/mL); (**c**) DNA concentration (μg/mL); (**d**) 16S rRNA-targeted qPCR. $R^2$ value for the 16S rRNA qPCR standard curve was 0.994. Blue lines are the species growth curves (*n* = 3, for qPCR *n* = 6).

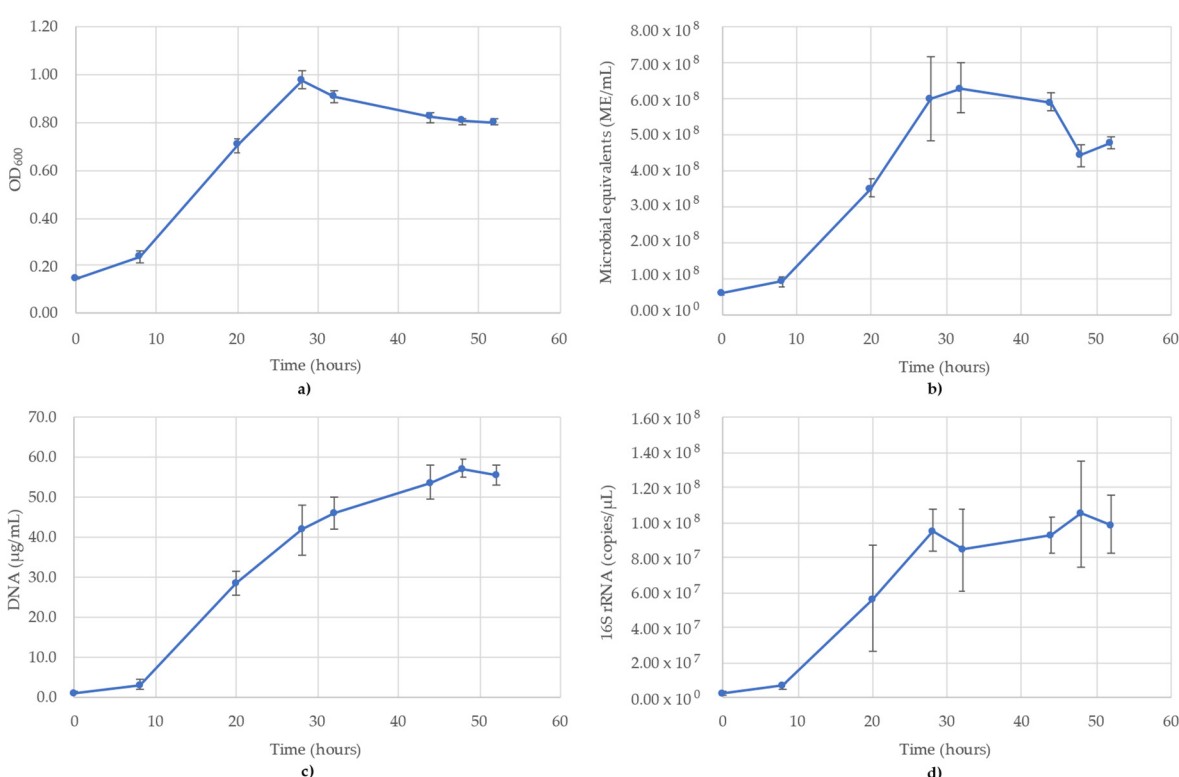

**Figure 6.** Monitoring of *T. aromatica* using (**a**) OD$_{600}$; (**b**) ATP (ME/mL); (**c**) DNA concentration (μg/mL); (**d**) 16S rRNA-targeted qPCR. $R^2$ value for the 16S rRNA qPCR standard curve was 0.979. Blue lines are the species growth curves (*n* = 3, for qPCR *n* = 6).

### 3.1. Acetobacterium Woodii Monitoring

The results of monitoring the *A. woodii* culture time course using $OD_{600}$, ATP (ME/mL), DNA (μg/mL) and 16S rRNA-targeted qPCR are shown in Figure 1. It is noted that the $OD_{600}$ values pass the value of 1.0, but for consistency of measuring the time course, no sample dilutions were performed (Figure 1a). The $OD_{600}$ values followed a sigmoidal growth curve over the 48 h monitored, though the stationary phase never reached a full plateau (Figure 1a). The ATP was similar to $OD_{600}$, though a clear stationary phase was observed between 32 and 48 h, where the values decreased slightly over this time (Figure 1b). The DNA concentrations also followed a sigmoidal curve, but peaked earlier (T = 24 h), which corresponds to the mid-log phase of the $OD_{600}$ and ATP trends. According to the DNA, the stationary phase began at 24 h (24.9 μg/mL), after which the concentration dipped slightly to 21.1 μg/mL (T = 28 h) and then remained stable between 23.6 μg/mL and 27.5 μg/mL (Figure 1c). The trends of the 16S rRNA qPCR results followed the DNA concentrations. The 16S rRNA copy numbers showed the same lag phase as DNA between 0 and 8 h and peaked at 24 h, before slightly increasing to $2.07 \times 10^7$ copies/μL by 48 h (Figure 1d). The $R^2$ value for the 16S rRNA standard curve used for *A. woodii* was 0.987.

### 3.2. Bacillus Subtilis Monitoring

The results of monitoring the *B. subtilis* culture time course using $OD_{600}$, ATP (ME/mL), DNA (μg/mL) and 16S rRNA-targeted qPCR are shown in Figure 2. The $OD_{600}$ showed what appears to be a diauxic growth curve, likely a result of the addition of oxygen to the headspace (Figure 2a). The ATP showed greater variability between replicates than the OD readings, reaching peak values at 13 h ($2.64 \times 10^8$ ME/mL) before dipping to $8.26 \times 10^7$ ME/mL at 24 h and subsequently recovering to $2.47 \times 10^8$ ME/mL at 37 h. After this point, the readings gradually decreased to $1.58 \times 10^8$ ME/mL at the final time point of 48 h (Figure 2b). DNA concentrations also showed a diauxic growth pattern, but showed the first peak at 24 h (Figure 2c) compared to the 13 h peaks seen in $OD_{600}$ and ATP (Figure 2a,b). The DNA concentrations then decreased until 37 h, before increasing again at 48 h. The 16S trends were very similar to the DNA concentrations and also peaked at 24 h ($1.11 \times 10^7$ copies/μL) and 48 h ($1.13 \times 10^7$ copies/μL) (Figure 2d). Time points from 13 h onward in the ATP, DNA and 16S all showed greater variance between replicates, minimizing the ability to speak on the shape of the curve; however, the mean values followed the same trends. The $R^2$ value for the 16S rRNA standard curve used for *B. subtilis* was 0.995.

### 3.3. Desulfovibrio Vulgaris Monitoring

The results of monitoring the *D. vulgaris* culture time course using $OD_{600}$, ATP (ME/mL), DNA (μg/mL) and 16S rRNA-targeted qPCR is shown in Figure 3. $OD_{600}$ showed a steady increase over the course of the 48 h testing period, rather than a sharp sigmoidal curve (Figure 3a). The cause of the significant $OD_{600}$ readings is the production of iron(II) sulfide precipitation resulting from the metabolism of *D. vulgaris* in the medium. The growth trend is muted, though still observable, despite initial readings at 1.494 and increasing to 2.066 by 52.5 h (Figure 3a). The ATP reading (ME/mL) did not follow a typical sigmoidal growth curve, but rather peaked at 31 h ($3.01 \times 10^8$ ME/mL) before declining to $1.56 \times 10^8$ ME/mL at 44 h, where it remained relatively stable for the remainder of the time points (Figure 3b). The DNA concentrations followed more of a sigmoidal curve compared to the $OD_{600}$ readings, owing to the high scattering properties of the media particulates, which minimized the effect of the cells alone in the $OD_{600}$, but was not seen with DNA concentrations. In the DNA concentrations, the lag phase was between 0 and 19 h (1.7–2.7 μg/mL), then increased until 31 h (28.7 μg/mL) and peaked at 44 h (37.0 μg/mL) (Figure 3c). The 16S rRNA followed the same trend as the DNA, though the slope was minimized by the significant error bars of the final time point owing to half the replicates (*n* = 3) reporting values an order of magnitude less than the other three replicates ($1.52 \times 10^8$ copies/μL $\pm 1.16 \times 10^8$). Prior to this final point, the 16S trend was identical

to DNA, peaking at 44 h then decreasing afterwards (Figure 3d). The $R^2$ value for the 16S rRNA standard curve used for *D. vulgaris* was 0.993.

### 3.4. Geoalkalibacter Subterraneus Monitoring

The results of monitoring the *G. subterraneus* culture time course using $OD_{600}$, ATP (ME/mL), DNA (µg/mL) and 16S rRNA-targeted qPCR are shown in Figure 4. $OD_{600}$ followed a typical sigmoidal growth curve, although the lag phase was less pronounced compared to the other monitoring methods. $OD_{600}$ peaked at 24 h ($OD_{600}$ = 0.329) before decreasing over the rest of the monitoring period (Figure 4a). The ATP did not follow a sigmoidal curve, but saw the peak occur at 12 h ($2.41 \times 10^8$ ME/mL) before decreasing to $6.30 \times 10^7$ ME/mL at the final time point (T = 36 h) (Figure 4b). The DNA concentrations followed a standard sigmoidal curve (Figure 4c), nearly identical to the qPCR curve. The distinction between $OD_{600}$ and DNA was a decrease following the peak at 24 h, indicating the decline of microbial biomass undetectable by $OD_{600}$. The lag phase occurred between 0 and 8 h, during which the DNA concentrations were between 4.2 and 6.1 µg/mL before increasing up to 32.7 µg/mL at 24 h and decreasing to 28.5 µg/mL at 32 h (Figure 4c). It should be noted DNA was not collected at the 36 h time point due to a lack of supplies; thus, the DNA concentrations could not be collected for the final time point. The 16S rRNA-targeted qPCR monitoring showed a lag phase between 0 and 8 h ($1.17 \times 10^7$–$1.72 \times 10^7$ copies/µL) before increasing to $5.38 \times 10^7$ copies/µL at 24 h then decreasing to $4.70 \times 10^7$ copies/µL at 32 h (Figure 4d). As with the DNA concentrations, no DNA was available for the final time point (T = 36 h) and therefore no values are reported. Of the four monitoring methods, $OD_{600}$, DNA and qPCR followed a sigmoidal curve (Figure 4a,c,d), while ATP followed more of a bell curve (Figure 4b). The $R^2$ value for the 16S rRNA standard curve used for *G. subterraneus* was 0.961.

### 3.5. Pseudomonas Putida Monitoring

The results of monitoring the *P. putida* culture time course using $OD_{600}$, ATP (ME/mL), DNA (µg/mL) and 16S rRNA-targeted qPCR are shown in Figure 5. $OD_{600}$, ATP and DNA monitoring all showed a sigmoidal growth curve with a short lag phase and an exaggerated stationary phase (Figure 5a–c). The $OD_{600}$ followed a typical sigmoidal growth curve that began its stationary phase after 24 h and peaked at 1.062 (T = 49.5 h) (Figure 5a). The ATP followed a very similar trend as $OD_{600}$, except there was no true stationary phase, as readings increased linearly from $5.77 \times 10^8$ ME/mL (T = 6 h) at the end of the logarithmic phase to $9.89 \times 10^8$ ME/mL at the final time point (T = 49.5 h, Figure 5b). The trends of the DNA concentrations matched the trends of $OD_{600}$ very closely for the duration of the monitoring, with a short lag phase (first 2 h), a logarithmic phase until 9 h (63.9 µg/mL) and then a stationary phase for the remainder (which still showed a small increase in DNA concentrations), reaching a maximum of 86.4 µg/mL DNA (Figure 5c). The 16S rRNA copies/µL show a far less tidy sigmoidal curve compared to the other monitoring methods. The lag phase still occurred between 0 and 2 h, before increasing to 6 h ($5.57 \times 10^7$ copies/µL) at the start of the stationary phase (Figure 5d). After T = 6 h, where other lines of evidence show a gradual increase into a plateau, the 16S rRNA data fluctuated between $3.36 \times 10^7$ and $7.20 \times 10^7$ copies/µL, with the peak values occurring at T = 24 h ($7.20 \times 10^7$ copies/µL). The $R^2$ value for the 16S rRNA standard curve used for *P. putida* was 0.994.

### 3.6. Thauera Aromatica Monitoring

The results of monitoring the *T. aromatica* culture time course using $OD_{600}$, ATP (ME/mL), DNA (µg/mL) and 16S rRNA-targeted qPCR is shown in Figure 6. All methods showed a short lag phase between 0 and 8 h, but the stationary phase showed variability between when it occurred. $OD_{600}$ showed a modified sigmoidal curve with a peak of 0.979 occurring immediately after the log phase (T = 24 h), then dropped to 0.821 at 44 h, where it plateaued until the final time point of 52 h (Figure 6a). ATP monitoring followed

a sigmoidal curve with a maximum value of $6.29 \times 10^8$ ME/mL at T = 32 h, where it remained until 44 h and then dropped to $4.41 \times 10^8$ ME/mL at T = 48 h (Figure 6b). The DNA concentrations followed a more gradual sigmoidal curve than ATP or $OD_{600}$, lacking a significant stationary phase (Figure 6c). The DNA concentrations peaked at 57.0 µg/mL at 48 h, after which there was a slight decline to 55.3 µg/mL at the last time point. The 16S rRNA copies/µL followed a trend most closely resembling $OD_{600}$, with a peak at 28 h ($9.57 \times 10^7$ copies/µL), after which there was a stationary phase until the final time point, during which time the values ranged between $1.27 \times 10^7$ and $1.50 \times 10^7$ (Figure 6d). The $R^2$ value for the 16S rRNA standard curve used for *T. aromatica* was 0.979.

### 3.7. Comparison of Cell Count Equivalents

To compare these methods directly, cell count equivalents were calculated for each method at time zero, a time point representative of mid-log phase and a time point representative of the stationary phase. The DNA was converted into a cell count proxy using the average DNA concentration per cell. Under typical sample conditions, it is impossible to calculate exact cell count equivalents from a complex environmental or unknown sample where the length of genomes is unknown and diverse. The comparison of the calculated cell count equivalents from the four methods are shown in Supplemental Data Table S2.

Time zero for all methods and species was on the order of $10^7$–$10^8$ cells/mL, with the exception of the ATP cell counts for *B. subtilis* (Table S2). The values for all methods and species increased between inoculation and mid-log phase data points, indicating that they all successfully measured an increase in cell counts. The DNA cell counts were typically the highest values out of the four methods, while $OD_{600}$ and ATP were the lowest values. The stationary phases for the DNA and 16S rRNA plateaued primarily at $10^{9-10}$ cells mL$^{-1}$ (*B. subtilis* 16S rRNA being the exception at $8.4 \times 10^8$ cells/mL). The ATP and $OD_{600}$ values plateaued at $10^{7-8}$ CFU/mL, with the single exception of the *A. woodii* ATP value ($1.45 \times 10^9$ ME/mL).

### 3.8. Mixed-Community Monitoring

A single trial of challenge to biocides THPS or BAC was run in a CDC reactor. Figure 7 shows the $OD_{600}$, ATP, DNA and 16S rRNA-targeted qPCR results from the planktonic and sessile growth of the THPS trial from parallel CBR: one receiving 37.5 ppm THPS and the other receiving no biocide, but undergoing the same flushing. Examining all four growth monitoring methods together, we can see certain trends emerging between different methods. Firstly, except for $OD_{600}$, the sessile values are lower than the planktonic cells, until day 14 when the values of the 16S rRNA become very similar. Interestingly, the DNA and 16S rRNA do not trend together as well as in pure-culture growth, as the planktonic cells (in CBR exposed to THPS) increased on day 10 according to the DNA, while the 16S rRNA showed a steady decline between days 7 and 14 in both planktonic samples. Observing a single sample type at a time, we see that all monitoring methods showed similar trends for both of the planktonic samples, though the degree of change between days was different (sometimes a greater increase, as in the case of the DNA readings for sessile cells after exposure to THPS compared to the ATP or 16S rRNA qPCR) (Figure 7b–d). Due to the number of samples and replicates, the 16S rRNA copy numbers had to be determined using two separate quantitative runs, where the $R^2$ values of the standard curves were 0.960 and 0.979.

Figure 8 shows the growth monitoring methods of the mixed community grown in parallel CBRs with a BAC exposure in one CBR. As with the THPS trial, the $OD_{600}$ readings of the planktonic and sessile cells are mixed, all ranging between 0.20 and 0.60 (Figure 8a). The ATP showed a distinct separation between planktonic and sessile cells. Planktonic activity increased on day 8 after BAC biocide was added to one CBR and fresh media added to both, and then decreased until day 14, after which the highest microbial activity was recorded on day 21 (Figure 8b). The DNA concentrations showed a near-continuous decrease from inoculation to the end of the experiment in the planktonic growth of the

BAC-exposed CBR, while the BAC-free planktonic cells decreased until day 14, and then showed a sharp increase on day 21 (Figure 8c). In this trial, the DNA and 16S rRNA showed the expected strong similarity in trends, with the greatest difference seen in the difference between the planktonic and sessile cells.

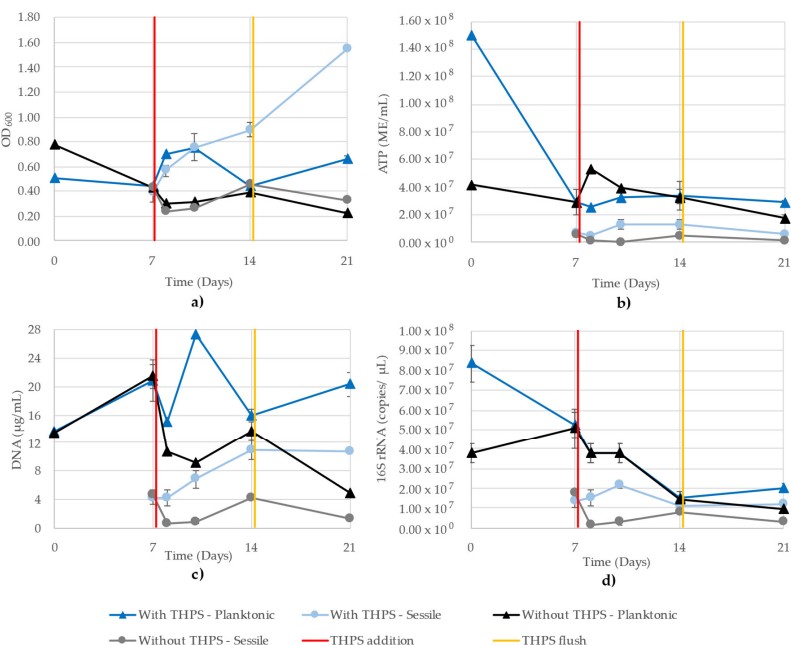

**Figure 7.** Planktonic and sessile growth monitoring of a mixed community grown in parallel CBR comprising *D. vulgaris*, *G. subterraneus*, *P. putida* and *T. aromatica* using (**a**) $OD_{600}$, (**b**) ATP (ME/mL), (**c**) DNA (μg/mL) and (**d**) 16S rRNA (copies/μL). $R^2$ values for the 16S rRNA qPCR standard curves were 0.960 and 0.979. One CBR received 37.5 ppm THPS on day 7 (red line) and both CBR were flushed on day 14 (yellow line).

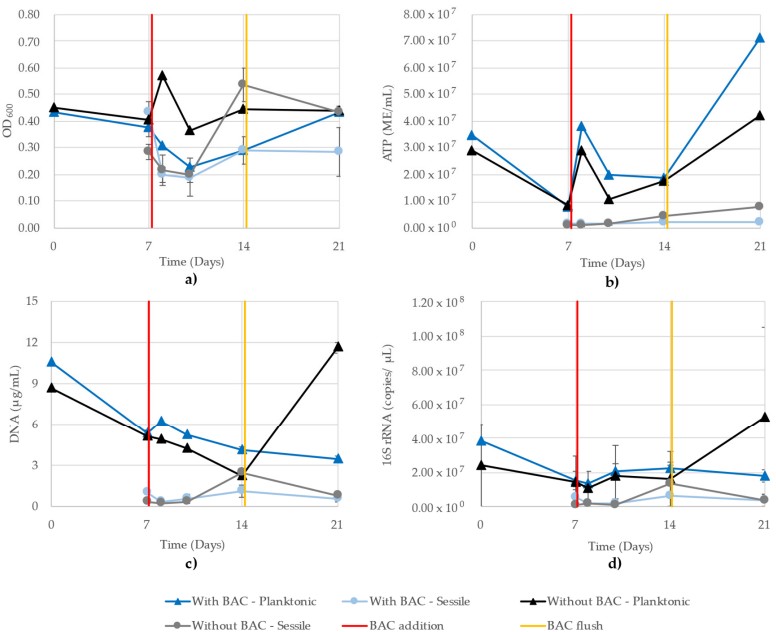

**Figure 8.** Planktonic and sessile growth monitoring of a mixed community grown in parallel CBR comprising *D. vulgaris*, *G. subterraneus*, *P. putida* and *T. aromatica* using (**a**) $OD_{600}$, (**b**) ATP (ME/mL), (**c**) DNA (μg/mL) and (**d**) 16S rRNA (copies/μL). $R^2$ values for the 16S rRNA qPCR standard curves were 0.971 and 0.993. One CBR received 1 ppm BAC on day 7 (red line) and both CBR were flushed on day 14 (yellow line).

All sessile cells were similar at days 7, 8 and 10, and then the BAC-free CBR cells increased on day 14 according to all monitoring methods (Figure 8). The $OD_{600}$ and ATP showed the difference was maintained between the two CBR sessile cells, while the DNA and 16S rRNA qPCR showed that the difference disappeared on day 21. Due to the number of samples and replicates, the 16S rRNA copy numbers had to be determined using two separate quantitative runs, where the $R^2$ values of the standard curves were 0.971 and 0.993.

## 4. Discussion

Here we briefly discuss how each individual method of monitoring microbial growth worked within each species. The trends in optical density nicely illustrated the expected sigmoidal growth curve in all species except *B. subtilis*, where a curve more similar to diauxic growth was observed. It is noted that *A. woodii*, *D. vulgaris* and *P. putida* all showed $OD_{600}$ measurements greater than 1.0. We recognize that OD data over 1 are not perfectly linearly related to the bacterial density. However, the focus of this work was to determine the level of correlation between the four monitoring methods (i.e., monitoring trends) and not the accuracy of the cell counts, so we have focused on the trends, which are still discernable even at these high $OD_{600}$ readings. Furthermore, as field samples would generate $OD_{600} > 1$, high OD readings accurately represent these types of samples, thus, we left our model cultures as they were. Despite the issues that high cell densities (or, in the case of *D. vulgaris*, precipitate) have on the transference of spectrophotometers, *A. woodii*, *D. vulgaris* and *P. putida* (all species with readings above 1.0) still showed the expected sigmoidal growth curve shape, albeit muted in the case of *D. vulgaris* (Figures 1a, 3a and 5a).

The ATP curves showed strong reproducibility between replicates during the lag phase in all species, but the replicates were less consistent during the mid-log phase (panel b, all figures). Compared to $OD_{600}$, the ATP readings showed a rapid decrease during the stationary phase, indicating that the death phase had started, but it was unable to be observed in $OD_{600}$. The DNA concentrations showed high reproducibility within replicates, apart from *B. subtilis* and, to a lesser extent, *P. putida*, possibly owing to the DNA extraction and recovery procedure. Although the species showed a sigmoidal curve, the amount of DNA recovered from each species varied, with the lowest amount being recovered from *B. subtilis* (12.0 $\pm$ 8.3 µg/mL at T = 48 h) and the highest amount from *P. putida* (86.4 $\pm$ 23.1 µg/mL at T = 49.5 h), while the rest were near 30–60 µg/mL (panel c, all figures).

Across the six pure-strain cultures, all four methods were able to measure the sigmoidal trends in biomass over time. In the case of *D. vulgaris* and *G. subterraneus* as measured by ATP, the trend was a bell curve and not sigmoidal growth, indicating that cellular activity in these species was highest during the mid-log phase and not steady throughout. The OD values were the least sensitive towards decreases in the late stages of the time courses, but typically had the lowest variability between replicates, and the 16S-targeted qPCR showed the highest variability (panel d, all figures). As expected, species with higher variability in DNA concentrations showed higher variability between replicates in the qPCR as well (i.e., *B. subtilis* and *P. putida*). The final time point of *D. vulgaris* showed high variability as a result of three of the six technical replicates being an order of magnitude above the others, but it is unclear what caused this.

Cell count equivalents from each of the testing methods were calculated into their unique cells $mL^{-1}$ (Table S2). The ATP microbial equivalents are an order of magnitude or two below the calculated DNA and 16S rRNA values, and have similar values as $OD_{600}$. In all methods, the change between the initial time point and the stationary phase is typically an order of magnitude regardless of the monitoring method. The lowest calculated cell count from all species along all time points is the microbial equivalent of *B. subtilis* at T = 0 h, which was three to four orders of magnitude below the other methods, but the discrepancy was closed by the mid-log phase (Table S2).

A side-by-side comparison of the cell count equivalents shows that none of the methods produced the same value for a time point. In many cases, different methods produced cell count equivalents on different orders of magnitude than the others. This indicates that following these calculations, the DNA and 16S rRNA calculations will likely overestimate cell counts compared to $OD_{600}$ and ATP.

To compare how the different methods agree with each other, scatter plots were used and linear correlation values were calculated (Supplemental Data Figures S3–S16). The R values were calculated for the full datasets of each monitoring method for each species and are reported in Table 6. From these values, we can see that the different methods have stronger correlations within certain species than others, such as *A. woodii*, which has a strong correlation between all methods (R = 0.92–1.00), while *B. subtilis* has a poor correlation across almost all methods (R = 0.64 to 0.89) except for DNA vs. 16S rRNA (R = 0.98).

The ATP measurements followed very similar trends to the $OD_{600}$ readings for all species (average R = 0.78, Table 6). The most marked differences occur in *D. vulgaris* (R = 0.65) and *G. subterraneus* (R = 0.24), where the shapes of the curves are significantly different owing to the peak in ATP occurring before the stationary phase. The strong correlation in the other four species supports the observation that the amount of ATP is consistent at all stages of the growth curve [45], while the *D. vulgaris* and *G. subterraneus* datasets disrupt the expected sigmoidal growth curve and show that the ATP concentrations are highest during the mid-log phase, indicating that this is not a universal rule. This is further exemplified in the *G. subterraneus* dataset, whose ATP values were markedly different from other approaches, and correspondingly, the linear correlations with ATP are all very low ($OD_{600}$ vs. ATP R = 0.24; ATP vs. DNA R = 0.16; ATP vs. 16S R = 0.15).

Unsurprisingly, DNA vs. 16S rRNA had a strong correlation (average R = 0.89), which is slightly skewed by the high average 16S rRNA copy number from the final time point of *D. vulgaris* (a result of the high variability between replicates), and with this final time point removed, the average R value improves to 0.95 (see Supplemental Data). The strongest correlation between any two methods is the $OD_{600}$ and DNA (average R = 0.94) and it follows that the correlation of 16S rRNA and $OD_{600}$ (average R = 0.87) would also be strong, with the higher average values in 16S rRNA replicates contributing to the average lower R value. The ATP vs. 16S rRNA methods had the lowest correlation on average between the six pure cultures, with an average R value of 0.65 (Table 6), which was mirrored in the correlation between DNA vs. ATP (average R = 0.72). However, the R values of these averages is skewed downwards as a result of the extremely poor values of *G. subterraneus*. With those values removed, the average R values improve to 0.84 for ATP vs. DNA (from 0.72) and 0.75 for ATP vs. 16S rRNA (from 0.65, see Supplemental Data). It is tempting to consider the R values with the omissions of the *G. subterraneus*; however, they provide a realistic comparison of the diversity of values one might expect in an environmental sample, even in such a small pool as the six species chosen here.

As this is a comparative study assessing four growth monitoring methods, we chose to use a mixed defined culture. This was deemed superior over environmental samples due to the lack of unknown factors that could lead to uninterpretable complexities confounding our ability to compare methods. The biocide challenges were conducted to determine the efficacy of these methods following a challenge to a community, and determine whether the methods could monitor fluctuations in growth. While this defined community is a fraction of the complexity of true environmental samples, this community reflects a reasonable level of complexity in a controlled manner to accurately assess these monitoring methods.

The inconsistencies observed in the $OD_{600}$ readings of the model community illustrate obvious issues with this technique when applied to samples with more than a pure, single-strain culture. Iron sulfide present in the media from *D. vulgaris* metabolism contributed to high readings in the CBR treated with THPS (Figure 7a). Alternatively, the CBR treated with BAC are a strong illustration of the insensitivity of the $OD_{600}$ assay, where the planktonic and sessile cells demonstrated similar values throughout the 21 days (Figure 8a). While

16S monitoring of the BAC-treated CBR showed similar values between planktonic and sessile cells on days 7 and 14, it maintained a distinction of planktonic cells showing greater values, which was observed in all other methods except for $OD_{600}$ (Figure 8d).

Unexpectedly, the addition of THPS did not have a negative impact on the mixed community, as seen in the steady increase in sessile cells following day 7, while the planktonic cells fluctuated (Figure 7). Similar trends were found between both BAC and THPS, where $OD_{600}$ showed similar values between planktonic and sessile growth (with the exception of THPS-exposed sessile cells on days 14 and 21; Figure 7a). The distinction between the planktonic and sessile cells seen in ATP, DNA and 16S rRNA qPCR indicates these testing methods are sensitive enough to distinguish between high and low cell mass.

Linear correlation values were determined for the four sample types collected from the parallel CBRs exposed to THPS and are reported in Table 7, and the CBRs exposed to BAC are reported in Table 8 (data used are provided in the Supplemental Data). The averages of all correlations from both trials are markedly lower than the averages seen in the pure culture correlations (Table 6). The correlation of the sessile cells without THPS have the highest correlations (R = 0.80–0.96), while the corresponding planktonic cells are lower (R = 0.11–0.66) (Table 7). The sessile cells exposed to THPS showed a very weak correlation between $OD_{600}$ and ATP (R = 0.04) and high values between $OD_{600}$ and DNA (R = 0.83), while the rest of the values were between 0.27 and 0.40. The planktonic cells exposed to THPS had a high correlation between ATP and 16S (R = 0.82), while the rest of the values were between 0.20 and 0.48.

The planktonic cells without BAC illustrate the difficulties with using $OD_{600}$ on field samples, as it showed a poor correlation with all other methods; contrastingly, ATP, DNA and 16S had strong correlations (Table 8). Conversely, the $OD_{600}$ correlated better with ATP, DNA and 16S in the sessile cells without BAC (R = 0.77, 0.89 and 0.88, respectively) than ATP correlated with either DNA (R = 0.48) or 16S (R = 0.46) (Table 8). All samples in the BAC CBR trial had a strong correlation between DNA and 16S (R = 0.76–0.96). Sessile cells exposed to BAC had a poor correlation of ATP with $OD_{600}$, DNA and 16S rRNA (R = 0.20–0.51), while $OD_{600}$, DNA and 16S all correlated well (R = 0.70–0.96) (Table 8).

In the mixed cultures, we can see the correlation values decrease and become less consistent compared to the pure cultures (which were all planktonic growth). In both CBR trials, the sessile cells not exposed to a biocide showed the strongest correlations, while the sessile cells exposed to biocide had lower values, yet the $OD_{600}$ to ATP was low in both.

These two model communities provide insight into how these monitoring methods work between the planktonic and sessile cells, as well as in response to a stress challenge. Here, two different biocides were used, but this serves as a proxy for a wide range of environmental challenges. These two trials provide valuable insight into situations that pure culture work cannot replicate, and illustrate that no single method is universally best for monitoring growth—although, as expected, $OD_{600}$ suffers the most in complex samples due to abiotic factors (in this case, iron sulfide precipitation) interfering with the readings. When considering field applications, the sessile samples are more applicable, as the majority of bacteria exist in sessile form [75–77]. Both of the biocide challenges reduced the correlation values of the sessile cells compared to the sessile cells without biocide (except for ATP to 16S and DNA to 16S of the BAC sessile cells).

## 5. Conclusions

This work set out to examine the monitoring techniques readily employed in the field and compare them to the methods best suited for lab cultures. $OD_{600}$ is poorly suited for field samples due to the requirement for a liquid medium and the presence of biological and non-biological materials frequently present in environmental samples (soil, aquatic/marine suspended sediments, wastewater flocculants, infection/wound material, plant materials, etc.) that will artificially increase $OD_{600}$ values. This is illustrated by the *D. vulgaris* dataset, where the scattering of light is increased due to the precipitation of iron sulfide resulting from sulfide production by *D. vulgaris*. Even without the presence of a precipitate in

the media, direct OD$_{600}$ comparisons between species offer little value, as the stationary phase values within these six species alone ranged from 0.317 to 1.296 (*G. subterraneus* and *A. woodii*, respectively). As a result, OD$_{600}$ is only suited to rapid monitoring of a pure culture, ideally one capable of being grown on agar to confirm CFU/mL values, and has no value in terms of cross-comparisons.

As shown in Figures 1–4, the trends are mostly consistent across the difference species. This indicates that while any single method (aside from OD$_{600}$) can be used with reasonable confidence to assess the microbial cell density in a particular system or environment, comparing multiple methods will lead to false assumptions regarding changes in cell concentrations.

Due to the need to include additional values for converting ATP, 16S rRNA copy numbers and DNA into cell count equivalents, it is more reasonable to leave these readings as their true output (i.e., pg ATP, copy number 16S rRNA and μg DNA, respectively) rather than adding a conversion factor and altering the output data. This is even more important when using environmental samples where biological factors (e.g., DNA amount per cell) will vary between species.

Applying these monitoring methods to a model community of four species with and without biocide challenge illustrated the variation that these methods may experience in more complex environments. Challenges such as the biocides used here were found to affect the resulting correlations, though correlations with DNA were, on average, higher than without DNA (Tables 7 and 8).

This work shows that using DNA concentrations as a proxy for cell counts could be considered the best universal indicator for microbial cell numbers. It carries a strong correlation to the OD$_{600}$ values of pure cultures in liquid media, is not as susceptible to large variation between replicates as qPCR, provides meaningful data without the use of a conversion calculation and can be used in downstream applications. The drawbacks of using DNA as a cell count proxy are the cost of extraction per replicate and the potential for issues in DNA recovery.

While this work focused on pure cultures, we believe these results can be extrapolated to mixed species and samples with highly diverse microbial populations. The simplest and most impactful conclusion from this work is that there is no universal or best method for monitoring microbial growth. It is more important to be consistent with a chosen monitoring technique and understand its limitations, as illustrated.

**Supplementary Materials:** The following supporting information can be downloaded at: https://www.mdpi.com/article/10.3390/microbiolres13020020/s1. Figure S1: Microbial equivalents per milliliter time course as determined using luciferase-based ATP assay for each species plotted on a log10 scale for: (**a**) *A. woodii*; (**b**) *B. subtilis*; (**c**) *D. vulgaris*; (**d**) *G. subterraneus*; (**e**) *P. putida*; (**f**) *T. aromatica*. Blue lines are the species growth curves (*n* = 3); Figure S2: 16S rRNA copies per microliter as detected by qPCR on a log10 scale over a time course of: (**a**) *A. woodii*; (**b**) *B. subtilis*; (**c**) *D. vulgaris*; (**d**) *G. subterraneus*; (**e**) *P. putida*; (**f**) *T. aromatica*. Blue lines are the species growth curves (*n* = 6, three biological each with two technical replicates); Figure S3: XY scatter plots of all four monitoring methods from *A. woodii* used to determine the correlation values (R values) between (**a**) OD$_{600}$ and ATP; (**b**) OD$_{600}$ and DNA; (**c**) OD$_{600}$ and 16S rRNA; (**d**) ATP and DNA; (**e**) ATP and 16S rRNA; (**f**) DNA and 16S rRNA. R values were calculated by taking the square root of the R$^2$ value from the linear trendlines; Figure S4: XY scatter plots of all four monitoring methods from *B. subtilis* used to determine the correlation values (R values) between (**a**) OD$_{600}$ and ATP; (**b**) OD$_{600}$ and DNA; (**c**) OD$_{600}$ and 16S rRNA; (**d**) ATP and DNA; (**e**) ATP and 16S rRNA; (**f**) DNA and 16S rRNA. R values were calculated by taking the square root of the R$^2$ value from the linear trendlines; Figure S5: XY scatter plots of all four monitoring methods from *D. vulgaris* used to determine the correlation values (R values) between (**a**) OD$_{600}$ and ATP; (**b**) OD$_{600}$ and DNA; (**c**) OD$_{600}$ and 16S rRNA; (**d**) ATP and DNA; (**e**) ATP and 16S rRNA; (**f**) DNA and 16S rRNA. Modified DNA and 16S rRNA correlation to exclude the final 16S (and corresponding DNA) time point is shown in orange. R values were calculated by taking the square root of the R$^2$ value from the linear trendlines; Figure S6: XY scatter plots of all four monitoring methods from *G. subterraneus* used to determine the correlation values

(R values) between (**a**) $OD_{600}$ and ATP; (**b**) $OD_{600}$ and DNA; (**c**) $OD_{600}$ and 16S rRNA; (**d**) ATP and DNA; (**e**) ATP and 16S rRNA; (**f**) DNA and 16S rRNA. R values were calculated by taking the square root of the $R^2$ value from the linear trendlines; Figure S7: XY scatter plots of all four monitoring methods from *P. putida* used to determine the correlation values (R values) between (**a**) $OD_{600}$ and ATP; (**b**) $OD_{600}$ and DNA; (**c**) $OD_{600}$ and 16S rRNA; (**d**) ATP and DNA; (**e**) ATP and 16S rRNA; (**f**) DNA and 16S rRNA. R values were calculated by taking the square root of the $R^2$ value from the linear trendlines; Figure S8: XY scatter plots of all four monitoring methods from *T. aromatica* used to determine the correlation values (R values) between (**a**) $OD_{600}$ and ATP; (**b**) $OD_{600}$ and DNA; (**c**) $OD_{600}$ and 16S rRNA; (**d**) ATP and DNA; (**e**) ATP and 16S rRNA; (**f**) DNA and 16S rRNA. R values were calculated by taking the square root of the $R^2$ value from the linear trendlines; Figure S9: XY scatter plots of all four monitoring methods from the planktonic cells from the CBR exposed to THPS used to determine the correlation values (R values) between (**a**) $OD_{600}$ and ATP; (**b**) $OD_{600}$ and DNA; (**c**) $OD_{600}$ and 16S rRNA; (**d**) ATP and DNA; (**e**) ATP and 16S rRNA; (**f**) DNA and 16S rRNA. R values were calculated by taking the square root of the $R^2$ value from the linear trendlines; Figure S10: XY scatter plots of all four monitoring methods from the planktonic cells from the CBR not exposed to THPS used to determine the correlation values (R values) between (**a**) $OD_{600}$ and ATP; (**b**) $OD_{600}$ and DNA; (**c**) $OD_{600}$ and 16S rRNA; (**d**) ATP and DNA; (**e**) ATP and 16S rRNA; (**f**) DNA and 16S rRNA. R values were calculated by taking the square root of the $R^2$ value from the linear trendlines; Figure S11: XY scatter plots of all four monitoring methods from the sessile cells from the CBR exposed to THPS used to determine the correlation values (R values) between (**a**) $OD_{600}$ and ATP; (**b**) $OD_{600}$ and DNA; (**c**) $OD_{600}$ and 16S rRNA; (**d**) ATP and DNA; (**e**) ATP and 16S rRNA; (**f**) DNA and 16S rRNA. R values were calculated by taking the square root of the $R^2$ value from the linear trendlines; Figure S12: XY scatter plots of all four monitoring methods from the sessile cells from the CBR not exposed to THPS used to determine the correlation values (R values) between (**a**) $OD_{600}$ and ATP; (**b**) $OD_{600}$ and DNA; (**c**) $OD_{600}$ and 16S rRNA; (**d**) ATP and DNA; (**e**) ATP and 16S rRNA; (**f**) DNA and 16S rRNA. R values were calculated by taking the square root of the $R^2$ value from the linear trendlines; Figure S13: XY scatter plots of all four monitoring methods from the planktonic cells from the CBR exposed to BAC used to determine the correlation values (R values) between (**a**) $OD_{600}$ and ATP; (**b**) $OD_{600}$ and DNA; (**c**) $OD_{600}$ and 16S rRNA; (**d**) ATP and DNA; (**e**) ATP and 16S rRNA; (**f**) DNA and 16S rRNA. R values were calculated by taking the square root of the $R^2$ value from the linear trendlines; Figure S14: XY scatter plots of all four monitoring methods from the planktonic cells from the CBR not exposed to BAC used to determine the correlation values (R values) between (**a**) $OD_{600}$ and ATP; (**b**) $OD_{600}$ and DNA; (**c**) $OD_{600}$ and 16S rRNA; (**d**) ATP and DNA; (**e**) ATP and 16S rRNA; (**f**) DNA and 16S rRNA. R values were calculated by taking the square root of the $R^2$ value from the linear trendlines; Figure S15: XY scatter plots of all four monitoring methods from the sessile cells from the CBR exposed to BAC used to determine the correlation values (R values) between (**a**) $OD_{600}$ and ATP; (**b**) $OD_{600}$ and DNA; (**c**) $OD_{600}$ and 16S rRNA; (**d**) ATP and DNA; (**e**) ATP and 16S rRNA; (**f**) DNA and 16S rRNA. R values were calculated by taking the square root of the $R^2$ value from the linear trendlines; Figure S16: XY scatter plots of all four monitoring methods from the sessile cells from the CBR not exposed to BAC used to determine the correlation values (R values) between (**a**) $OD_{600}$ and ATP; (**b**) $OD_{600}$ and DNA; (**c**) $OD_{600}$ and 16S rRNA; (**d**) ATP and DNA; (**e**) ATP and 16S rRNA; (**f**) DNA and 16S rRNA. R values were calculated by taking the square root of the $R^2$ value from the linear trendlines. Table S1: Tabulated data of $OD_{600}$, ATP and DNA methods used to determine proper sampling time points; Table S2: Summary of the calculated cell counts per milliliter from each monitoring method at the initial time point, mid log phase and stationary phase for all species tested.

**Author Contributions:** Conceptualization, D.C.B. and R.J.T.; methodology, D.C.B.; investigation, D.C.B.; resources, R.J.T.; writing—original draft preparation, D.C.B.; writing—review and editing, R.J.T.; supervision, R.J.T.; project administration, R.J.T. All authors have read and agreed to the published version of the manuscript.

**Funding:** The work was supported by Genome Canada through a Large Scale Applied Research Project (LSARP) grant, grant number 10202. DCB was supported with a PhD scholarship from the Natural Sciences Engineering Research Council of Canada (NSERC), funding reference number CGS-D2-535222-2019. RJT was also supported by an NSERC Discovery grant, grant number RGPPIN/04811-2015.

**Institutional Review Board Statement:** Not applicable.

**Informed Consent Statement:** Not applicable.

**Data Availability Statement:** Not applicable.

**Acknowledgments:** Special thank you to LuminUltra and Promega for contributing ATP and qPCR reagents as in-kind contributions to this work.

**Conflicts of Interest:** The authors declare no conflict of interest.

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
