# Peer review of "Assessing Microbial Monitoring Methods for Challenging Environmental Strains and Cultures"

_2036-7481, doi:10.3390/microbiolres13020020_

Round 1
Reviewer 1 Report
This is a problem commonly acknowledged by workers who require microbial counts and so is very relevant, but not original. Nevertheless, I think it is well worth publishing, as no similar article including comparisons with and between nucleic acid measurements is available. The article is very well written, especially the Introduction. Some minor errors of English are present, especially, in the Methods section (please note that "media" is a plural noun). It is important that future studies include real-life samples, with mixed populations, and I think the authors might fruitfully expand on this in their discussion.
Author Response
A section on a mixed model community of four species grown in a biofilm reactor has been added. The choice of using a model community over an environmental sample was made to avoid complications from the complexity of environmental samples. We believe the application of these methods to this model community demonstrates how these methods work in more complicated systems.
Reviewer 2 Report
The manuscript presents a comparison of different analytical methods to measure cell count or biomass in microbiological samples. The paper fits into the scope of the Journal and presents exemplarily the deviation, strengths and weaknesses of different analytical methods. The presented experiments are exclusively from pure cell cultures but are to be applied to environmental samples. A standard reference method for the comparison is missing. The text is written and structured well. There was no supplement available for the review of the paper.
In principle, the publication should be verified with reference measurements and the evaluation should be more focused. The paper should be accepted after mayor revision. The following comments might help to improve the paper:
Title
Not the growth (deviation or change over time) was monitored; the cell count or biomass was measured.
Material und Methoden
A standard reference method like CFU, cell counts or biomass (wet or dry weight) is missing. The validation of the method on environmental samples is missing.
Page 5, line 191-195: There is no reference for the cultivations given. Therefore the question arises if the preculture was incubated for 7 days without any passaging? In this case, it would be doubtful that the cells are in exponential growth for inoculation. During the cultivation, the steady state is reached after 40-60 h (2-3 days).
Page 5, line 191-195: What means 10% inoculant (what is 100%)? On which cell density, cell count or optical density was inoculated?
Page 7, Eq (1): As correctly noted, the calibration would have to be done separately for each cell line. Simply leave OD as OD.
Page 7, Eq (2): What is blankRLU, what is standardRLU?
Results
Page 8, figure 1: The orange line is OD of inoculant. If 10% of inoculant was used for inoculation and diluted in the same volume, the OD should be round about 10% of the inoculant in the beginning. This is not the case, please discuss. Furthermore, the orange line or inoculant OD is never mentioned in the text.
Page 8, Figure 1c: Why is the OD of this cell line at the beginning so high? Please describe and discuss.
Figure 1 to 4: Maybe it would be better to show the four courses OD, ME, DNA and RNA for each cell line in one graph, then similar courses can be easily recognized. Describe and discuss the courses by cell line and not by method.
Figure 4: Most RNA measurements show a large dispersion. Is that biological variance or measurement noise?
Table 6: unnecessary because, as can be seen from the equation on page 7, the values for each cell line are proportional to each other. The more sensible of the two should be chosen.
Table 7: superfluous if they are presented together in the figures. Furthermore, averaging over a process phase is not helpful for this purpose.
Table 8: No supplement was available for the review of the paper. A brief description of the procedure would have been helpful (how many samples, paired or averaged). Why is R² used for correlation instead the correlation coefficient R?
Author Response
Reviewer 2.
The manuscript presents a comparison of different analytical methods to measure cell count or biomass in microbiological samples. The paper fits into the scope of the Journal and presents exemplarily the deviation, strengths and weaknesses of different analytical methods. The presented experiments are exclusively from pure cell cultures but are to be applied to environmental samples. A standard reference method for the comparison is missing. The text is written and structured well. There was no supplement available for the review of the paper.
Reply - See reply to Reviewer 1 regarding environmental samples. The data for the range finding experiments have been added as Appendix B and included in the main manuscript.
In principle, the publication should be verified with reference measurements and the evaluation should be more focused. The paper should be accepted after mayor revision. The following comments might help to improve the paper:
Title
Not the growth (deviation or change over time) was monitored; the cell count or biomass was measured.
Reply - We appreciate this comment as it is a conversation of debate. Biomass may be a good alternative, but it also considers live, dead and EPS materials. Viable Cell counts is not considered possible with most bacteria and thus the ATP proxy which is metabolism rates not necessarily cell numbers, DNA levels captures dead cells as well, This is highlighted throughout the manuscript. However, the title and abstract has been altered to focus on the concept of ‘monitoring’ microbial cell biomass/biomolecule/metabolic activity changes with time as proxy for the traditional CFU growth curve.
Material und Method
Reply - A standard reference method like CFU, cell counts or biomass (wet or dry weight) is missing. The validation of the method on environmental samples is missing.
CFU is not considered an appropriate option for most bacteria due to the unculturable issues. CFU is also rarely used in non-health care industrial settings. Thus, comparison to CFU is irrelevant here.
Page 5, line 191-195: There is no reference for the cultivations given. Therefore the question arises if the preculture was incubated for 7 days without any passaging? In this case, it would be doubtful that the cells are in exponential growth for inoculation. During the cultivation, the steady state is reached after 40-60 h (2-3 days).
Reply - The cultures were incubated for seven days after freezer recovery into 20 mL of fresh medium. The first passage occurred on day 7, grown for another 7 days then the second passage was performed, which was the initiation of the study at time zero. This same treatment was performed during the initial trials (added as an additional Appendix B instead of supplemental data). It is acknowledged the cultures were not in mid log phase at the time of inoculation, but the sampling time points were chosen to reflect the recovery of a stationary phase culture.
Page 5, line 191-195: What means 10% inoculant (what is 100%)? On which cell density, cell count or optical density was inoculated?
Reply - Inoculation was 10% by volume of a culture from stationary phase. Clarification was added in text.
Page 7, Eq (1): As correctly noted, the calibration would have to be done separately for each cell line. Simply leave OD as OD.
Reply - Thank you, we have clarified this equation and left as OD.
Page 7, Eq (2): What is blankRLU, what is standardRLU?
Reply - Clarified in text.
Results
Page 8, figure 1: The orange line is OD of inoculant. If 10% of inoculant was used for inoculation and diluted in the same volume, the OD should be round about 10% of the inoculant in the beginning. This is not the case, please discuss. Furthermore, the orange line or inoculant OD is never mentioned in the text.
Reply -
Reply – Sorry this is distracting and misleading, the orange line has been removed in all figures.
Page 8, Figure 1c: Why is the OD of this cell line at the beginning so high? Please describe and discuss.
Reply - This is due to the dark precipitate produced as a result of the metabolism of D. vulgaris. This has been addressed in the text.
Figure 1 to 4: Maybe it would be better to show the four courses OD, ME, DNA and RNA for each cell line in one graph, then similar courses can be easily recognized. Describe and discuss the courses by cell line and not by method.
Reply - We wanted to highlight the differences of the unique methods across the different species. We feel changing to the four methods for each species makes the comparison of the methods as it functions between species more difficult and detracts from the discussion.
Figure 4: Most RNA measurements show a large dispersion. Is that biological variance or measurement noise?
Reply - This is an amplification of the differences seen in the DNA concentrations, though in some cases it was instrument noise. A sentence has been added to address this in the text.
Table 6: unnecessary because, as can be seen from the equation on page 7, the values for each cell line are proportional to each other. The more sensible of the two should be chosen.
Reply - Removed.
Table 7: superfluous if they are presented together in the figures. Furthermore, averaging over a process phase is not helpful for this purpose.
Reply - Table 7 has been moved to supplementary. We still believe it is a useful tool for comparison where all the data is in the same units.
Table 8: No supplement was available for the review of the paper. A brief description of the procedure would have been helpful (how many samples, paired or averaged). Why is R² used for correlation instead the correlation coefficient R?
Reply - Unclear why you could not see the supplementary as it was uploaded. This has been corrected to the use of R. A brief explanation of how it has been calculated has been included.
Reviewer 3 Report
line 19-21 of the abstract: What other three assays?
I found the intro too long.
Were the CFUs calculated or were they plated? it was not very clear
From this conclusion, it would be very interesting to focus on a single microorganism and then compare it to its pure culture as it has been done so far, with a sample in the field, with real conditions. And then determine the best form of monitoring in a real situation. The work was well analyzed, well discussed, but it concludes what we expected, that each case is a case, and there are particularities in each case. If they carry out at least one analysis with real conditions, I believe it would add much more to the work
Author Response
Reviewer 3.
line 19-21 of the abstract: What other three assays?
Reply - Clarified.
I found the intro too long.
Reply - The introduction was written as a mini review of the methods typically used to help guide any readers on which methods would best suit their needs. We feel this overview is very important to our paper, as a source to readers as well to put the readers in context to our study.
Were the CFUs calculated or were they plated? it was not very clear
Reply - They were calculated. This has been clarified in text. this was generated as a way to compare with similar units. The related Table is now placed in the supplementary.
From this conclusion, it would be very interesting to focus on a single microorganism and then compare it to its pure culture as it has been done so far, with a sample in the field, with real conditions. And then determine the best form of monitoring in a real situation. The work was well analyzed, well discussed, but it concludes what we expected, that each case is a case, and there are particularities in each case. If they carry out at least one analysis with real conditions, I believe it would add much more to the work
Reply - An analysis of a mixed model community monitored with the four methods used has been added. As mentioned in response to Reviewer 1, we believe this provides a meaningful insight into how these methods report biomass in a controller manner.
Round 2
Reviewer 3 Report
The authors answered all the points raised. I believe the article is ready for publication
Author Response
Response to reviewers
Assessing microbial monitoring methods for challenging environmental strains and cultures (microbiolres-1673657)
Extensive changes to the results section were made in response to the suggestion of reviewers 2 and 3. The subsections of the results have been changed to reflect the single species monitoring as opposed to the monitoring methods. Figures Have been adjusted accordingly, now there are six figures of the pure culture strains, each with four panels, one for each monitoring method. The numbered subsections for “Comparison of cell count equivalents” and “Mixed community monitoring” have been changed accordingly to 3.7 and 3.8 (previously 3.4 and 3.5).
A brief discussion of the trends between monitoring methods of the pure cultures was added to the Discussion section. This addition is on lines 825 – 832 of the track changes document.
The format of chemicals and devices used in the experiment have been corrected as per suggestions. These changes were made in lines 215-224, 230, 238, 240, 299 and 300 of the track changes document.
Numbering of the Supplemental data within the text have been corrected to be consistent throughout. Appendices have all been removed from the end of the document and moved into the supplemental data.